# Development of a Multiplex Tandem PCR (MT-PCR) Assay for the Detection of Emerging SARS-CoV-2 Variants

**DOI:** 10.3390/v13102028

**Published:** 2021-10-08

**Authors:** Richard Hale, Peter Crowley, Samir Dervisevic, Lindsay Coupland, Penelope R. Cliff, Saidat Ebie, Luke B. Snell, Joel Paul, Cheryl Williams, Paul Randell, Marcus Pond, Keith Stanley

**Affiliations:** 1AusDiagnostics UK Ltd., Unit 3 Anglo Business Park, Chesham HP5 2QA, UK; 2AusDiagnostics Pty Ltd., Mascot, Sydney, NSW 2020, Australia; peter.crowley@ausdx.com (P.C.); keith.stanley@ausdx.com (K.S.); 3NRP Innovation Centre, Microbiology Department, Norfolk and Norwich University Hospital, Norwich Research Park, Norwich NR4 7GJ, UK; samir.dervisevic@nnuh.nhs.uk (S.D.); lindsay.coupland@nnuh.nhs.uk (L.C.); 4Infection Sciences, St Thomas’ Hospital, London SE1 7EH, UK; penny.cliff@viapath.co.uk (P.R.C.); saidat.ebie@gstt.nhs.uk (S.E.); luke.snell@gstt.nhs.uk (L.B.S.); 5Department of Microbiology, The Royal Oldham Hospital, Oldham OL1 2JH, UK; joel.paul@pat.nhs.uk (J.P.); cheryl.williams2@pat.nhs.uk (C.W.); 6Department of Infection and Immunity, North West London Pathology, Charing Cross Hospital, London W6 8RF, UK; paulrandell@nhs.net (P.R.); marcus.pond@nhs.net (M.P.)

**Keywords:** SARS-CoV-2, variants, in vitro diagnostic test

## Abstract

The emergence of variants of SARS-CoV-2 has created challenges for the testing infrastructure. Although large-scale genome sequencing of SARS-CoV-2 has facilitated hospital and public health responses, access to sequencing facilities globally is variable and turnaround times can be significant, so there is a requirement for rapid and cost-effective alternatives. Applying a polymerase chain reaction (PCR)-based single nucleotide polymorphism (SNP) approach enables rapid (<4 h) identification of SARS-CoV-2 lineages from nucleic acid extracts, through the presence or absence of a panel of defined of genomic polymorphisms. For example, the B.1.1.7 lineage (“UK”, “Alpha”, or “Kent” variant) is characterised by 23 mutations compared to the reference strain, and the most biologically significant of these are found in the S gene. We have developed a SARS-CoV-2 typing assay focused on five positions in the S gene (HV69/70, N501, K417, E484 and P681). This configuration can identify a range of variants, including all the “Variants of Concern” currently designated by national and international public health bodies. The panel has been evaluated using a range of clinical isolates and standardised control materials at four UK hospitals and shows excellent concordance with the known lineage information derived from full sequence analysis. The assay has a turnaround time of about three hours for a set of up to 24 samples and has been utilised to identify emerging variants in a clinical setting.

## 1. Introduction

In December 2019, a novel coronavirus, severe acute respiratory syndrome coronavirus 2 (SARS-CoV-2), was first isolated from Wuhan city, China and within three months of this discovery, the global community was challenged with a devastating pandemic [1].

Whole-genome sequencing provides high-resolution data that enable investigation of pathogen evolution and population structure. When combined with robust epidemiological data, it is possible to gain insights into SARS-CoV-2’s origins, transmission and responses to control measures. Since the start of the pandemic, sequencing efforts and data sharing have facilitated tracking of the pandemic, identifying multiple independent virus introductions into different countries and novel mutations in the SARS-CoV-2 genome.

Whilst tracking genetic variations from positive SARS-CoV-2 samples yields crucial information about the number of variants circulating in an outbreak and the possible lines of transmission, sequencing every positive SARS-CoV-2 sample would be prohibitively costly for population-scale test and trace operations [2]. PCR-based genotyping is a rapid, high-throughput and cost-effective alternative for screening positive SARS-CoV-2 samples in many settings.

In July 2021, the UK Government issued guidance on the mutations required for reflex testing for genotyping [3]. This specifies detection of P681R, K417T, K417N and E484K as the key mutations to infer Beta, Gamma, Delta, Kappa and VUI-21APR-03 variants. These polymorphisms in the SARS-CoV-2 spike gene are reported to have important roles in viral infectivity, transmissibility and potentially vaccine escape [4,5]. The UK Government guidance was followed in the design of a multiplex-tandem PCR assay (MT-PCR, [6]) with the further addition of N501Y to include detection of the Alpha variant.

The first step of this MT-PCR assay is highly multiplexed, containing eight pairs of outer primers for the target genes (together with a pair of primers for an internal control and a human sample adequacy gene), enabling rapid pre-amplification of all target amplicons simultaneously. The reaction mixture includes reverse transcriptase to enable amplification of viral RNA, but only 15 cycles of PCR are carried out. This low cycle number minimises competition and maintains the relative concentration of the different amplicons. After the initial amplification, each sample is diluted and dispensed into 16 separate “Step 2” reaction wells, which contain the inner allele-specific primers, and these are amplified for 30 cycles using real time PCR. Wild type and variant-specific Step 2 primers differ at their 3′ ends, resulting in approximately 10 cycles’ difference in Ct, which allows clear identification of wild-type or mutation. Following the Step 2 amplification, data from the amplicon melts are used to confirm that they are within the expected range.

This manuscript details a technical performance assessment of the SARS-CoV-2 Variant Typing Panel (16-well), performed across a network of virology laboratories, to enable identification of variants of concern (VOC) currently designated by Public Health England.

## 2. Materials and Methods

### 2.1. Assay Design

The AusDiagnostics High-Plex SARS-CoV-2 (16-well) panel (Ref. 20082 Ver. 06), AusDiagnostics UK Ltd., Buckinghamshire, United Kingdom, was designed to use a diagnostic algorithm to predict the genotype present at five positions in the SARS-CoV-2 surface glycoprotein (S gene) (501, 69–70, 417, 484, and 681) associated with SARS-CoV-2 VOC (Variants of Concern). An additional three SARS-CoV-2-specific assays targeting ORF1, ORF6 and ORF8 were included for confirmation of the presence of SARS-CoV-2 within the sample. For each position, the cycle threshold (Ct) of an assay specific to the wild type (_WT_) at that position and the Ct of an assay specific mutant are first compared against the internal reference to obtain a measure of concentration. The concentration ratio of each variant pair is then calculated, and a diagnostic algorithm is applied to predict the genotype present. Where SARS-CoV-2 is present in the sample, there are two possible results for the 501 (N501_WT_ or N501Y detected) and 69–70 (HV69–70del absent/present) positions. There are three possible results for the 417 (K417_WT_, K417N detected or K417T detected), 484 (E484_WT_, E484K detected or E484Q detected) and 681 (P681_WT_, P681R detected or P681 other mutation detected) positions. Results for each variant are displayed via the AusDiagnostics Results software in the “Diagnosis” box (Figure 1). If the sample is SARS-CoV-2-negative, no typing result is displayed.

### 2.2. Clinical Specimens and other Material

The diagnostic sensitivity and specificity were assessed using: (a) 82 residual clinical respiratory samples, (b) Nucleic Acid Amplification Techniques (NAT) Panel for SARS-CoV-2 (20/266) (National Institute for Biological Standards and Controls (NIBSC), Hertfordshire, UK) and (c) NATtrol™ Respiratory Pathogen Panel-1 (Zeptometrix Corporation, New York, United States). The analytical sensitivity of the SARS-CoV-2 Variant Typing Panel (16-well) was performed using a dilution series of the first WHO International Standard for SARS-CoV-2 RNA (20/146) for Nucleic acid Amplification Technique (NAT)-based assays consisting of the acid-heat inactivated England/02/2020 isolate of SARS-CoV-2 (NIBSC, Hertfordshire, UK), as well as Alpha (VOC-20DEC-01, B.1.1.7), Beta (VOC-20DEC-02, B.1.351), Gamma (VOC-21JAN-02, P.1) and Delta (VOC-21APR-02, B.1.617.2) variant material provided by PHE Porton. The first WHO International Standard for SARS-CoV-2 RNA (20/146) was reconstituted in 0.5 mL of molecular-grade water to provide a final concentration of 7.70 Log10 IU/mL prior to preparation of the dilution series. The dilution series of the first WHO International Standard for SARS-CoV-2 RNA (20/146) as well as the Alpha (VOC-20DEC-01, B.1.1.7), Beta (VOC-20DEC-02, B.1.351), Gamma (VOC-21JAN-02, P.1) and Delta (VOC-21APR-02, B.1.617.2) variant material were prepared in L6 Lysis Buffer (Severn Biotech, Worcestershire, UK). The estimated sensitivity and specificity of the SARS-CoV-2 Variant Typing Panel (16-well) was calculated as 100% X (TP/(TP + FN)) and 100% X (TN/(TN + FP)), respectively.

### 2.3. Nucleic Acid Extraction for the Validation Process

Total nucleic acid was extracted from 200 µL of SARS-CoV-2-positive archived clinical respiratory samples and eluted into a final volume of 60 µL of elution buffer using the QIAsymphony^®^ DSP Virus/Pathogen Mini Kit (QIAGEN, Manchester, UK) on the QIAsymphony^®^ SP instrument (QIAGEN, UK), according to the manufacturer’s instructions. In total, 10 µL of extract were used for analysis with the typing panel. Detection of an artificial sequence (SPIKE) was included as a control to ensure the integrity of the reagents, equipment function and the presence of inhibitors in the sample.

### 2.4. Nucleic Acid Sequencing

Nucleic acid sequencing was carried out through either the COG-UK consortium [7] or using in-house methods (St Thomas’ Hospital) where whole-genome sequencing of residual samples from SARS-CoV-2 cases was performed on GridION (Oxford Nanopore Technology), using version 3 of the ARTIC protocol and bioinformatics pipeline [8], with lineage calling by pangolin [9].

## 3. Results

### 3.1. Assay Sensitivity, Specificity, and Limits of Detection

The analytical specificity of the panel with respect to cross-reactivity with respiratory pathogens other than SARS-CoV-2 was assessed using the NAT Panel for SARS-CoV-2 (20/266) (NIBSC, UK), which included samples positive for Coronaviruses 229E and NL63, RSV and Influenza virus B (B/Jiangsu/10/2003) as well as the NATtrol™ Respiratory Pathogen Panel-1 (five pools of viral and bacterial respiratory pathogens and a negative control). There was no cross-reactivity observed with the viral or bacterial pathogens. The analytical specificity for variant detection was determined using the first WHO International Standard for SARS-CoV-2 RNA (20/146) for NAT-based assays consisting of acid-heat inactivated wild-type England/02/2020 isolate of SARS-CoV-2 and the NAT Panel for SARS-CoV-2 (20/266), which consisted of a panel of 24 samples that were positive for acetic acid and heat-inactivated wild-type SARS-CoV-2 virus Melbourne strain (BetaCoV/Australia/VIC01/2020). There was no non-specific cross-reactivity for SARS-CoV-2 E484K, K417N, K417T or P681R mutations detected. The diagnostic sensitivity of the panel was determined using a panel of 24 SARS-CoV-2-positive archived clinical respiratory samples previously characterised by whole genome sequencing (WGS) as the following lineages: B.1.1.7 (*n* = 12), B.1.351 (*n* = 4), B.1.1.248 (P.2) (*n* = 1), B.1.617.2 (*n* =7) and B.1.177 (*n* = 63). The “wild type” E484 and K417 targets were detected in all the samples of the B.1.177 lineage. Specificity and sensitivity for the mutations HV69/70, N501Y, K417N, E484K and P681R are shown in Table 1. Calculation of the estimated sensitivity and specificity for the target mutations was limited by the availability of samples of the specific lineages and was not possible at all for K417T, as this mutation was not present in any of the lineages available.

The analytical sensitivity (limit of detection) of the panel for the mutations K417N, K417T, E484N and P681R was determined using a tenfold dilution series of wild type and variant material (Table 2).

Readout from the panel was via the AusDiagnostics Results software and the diagnostic algorithm providing the “call” at each position (Figure 1). Assay turnaround time (TAT) was found to be less than 24 h (from extract to reported result).

### 3.2. Identification of a New SNP Combinations

Genotyping at St Thomas Hospital of two clinical samples from southeast London, identified as positive at St Thomas Hospital on 10th and 14th July 2021, gave an unexpected pattern of SNPs: N501Y, HV69/70_WT_, K417N, E484K and P681H (other). This sample was further analysed by sequencing and identified as pangolin lineage B.1.621 (“Colombian”) with 100% certainty. However, the SNP “K417N” was also confirmed (G22814T, >95% of >3000 reads), which is not currently associated with this lineage. It is therefore likely that this represents a new variant [10], analogous to the situation with the so-called Delta Plus variant, which also has K417N as an extra mutation [11].

Genotyping of a clinical sample (from a male patient with diffuse large B-cell lymphoma admitted with fever) at Norwich and Norfolk Hospital was called as N501Y, HV 69/70 deletion, K417T, E484_WT_ and P681H (other). Genome sequencing confirmed the P681H mutation but much of the Spike gene was missing from sequence analysis, so further confirmation was not possible. This sample appears to be of Alpha lineage (T716I and D1118H in the S gene were identified from the partial sequence data available), but with K417T, which has previously only been associated with the Gamma lineage.

Genotyping of a clinical sample identified as positive at Charing Cross Hospital in July was called as N501Y, HV69/70 _WT_, K417N, E484K and P681R. This sample failed to return a genome sequence due to technical error; therefore, it was not possible to assign a pangolin lineage.

### 3.3. Identification of a Novel ORF8 Deletion

A small cluster of samples were identified at Oldham Hospital in January 2021, where the ORF8 gene was absent, whereas the ORF1 gene and sample adequacy genes were detected well. The cluster was subsequently investigated further using Sanger sequencing and was found to carry a previously unknown 394 nucleotide deletion in the ORF8 gene (Figure 2) together with a D3L mutation in the N-gene. Analysis using the typing panel confirmed the absence of ORF8 and identified the genotype to be most likely derived from the “Alpha” lineage: N501Y, HV 69/70 deletion, K417_WT_, E484_WT_ and P681H (other).

## 4. Discussion

The typing panel described in this manuscript has been successfully deployed as a “reflex” assay to aid the identification of designated SARS-CoV-2 variants of concern (VOC). The assay has demonstrated the high sensitivity and selectivity as would be expected from a PCR-based approach and the TAT is significantly shorter (<4 h) than typically achieved with WGS (>24 h). Underpinning the requirement for the implementation of this technique is a requirement to describe the population structure of SARS-CoV-2 lineage attributable to both hospital and community infection within the United Kingdom.

In addition to yielded data for population-level epidemiological analysis, genotyping has enabled enhanced patient-level management in specific circumstances. For example, genotyping has enabled identification of patients demonstrating long-term infection with SARS-CoV-2, through the repeat identification of mutations associated with lineages that are currently not circulating within the community.

Results from clinical use of this panel have shown that it can identify emergent variants (or rather, combinations of SNPs), which may flag these infections for rapid infection control and public health interventions. The format of the assay, where both the “wild-type” and variant SNP are measured simultaneously, has the potential to allow early identification of other SNPs, such as E484A (a potential “escape” mutation [12]), as it will exhibit delayed amplification (typically 6–10 cycles) in both assays. Automated melt curve analysis is also provided for all the amplicons and this has the potential to give further information about a variant [13]. The selection of the SNPs for this panel was strongly predicated on the guidance of the UK Government, but the same approach could, in principle, be made for any set of SNPs/deletions. As these arise, the panel design can be modified and deployed rapidly for use on the AusDiagnostics High-Plex platform. An updated panel might also include assays for the detection of T478K (associated with inter alia the B.1.1.159 lineage and delta variant), L452R (delta, kappa, and a number of other lineages), L452Q (lambda), E1092K (theta) and so on [14]. Assay choice would be based on national guidance, prevalence, clinical and epidemiological significance and, in the High-Plex format, would be limited to 23 individual assays.

PCR-based genotyping methodologies will not replace genome sequence as the reference methodology for SARS-CoV-2 lineage assignment and epidemiological analysis. However, as national public health bodies work to implement genome-based pathogen surveillance, the reduced cost, favourable TAT and ease of operational deployment of genotyping assays should be recognised as part of the armamentarium of current and future pandemic response.

## Figures and Tables

**Figure 1 viruses-13-02028-f001:**
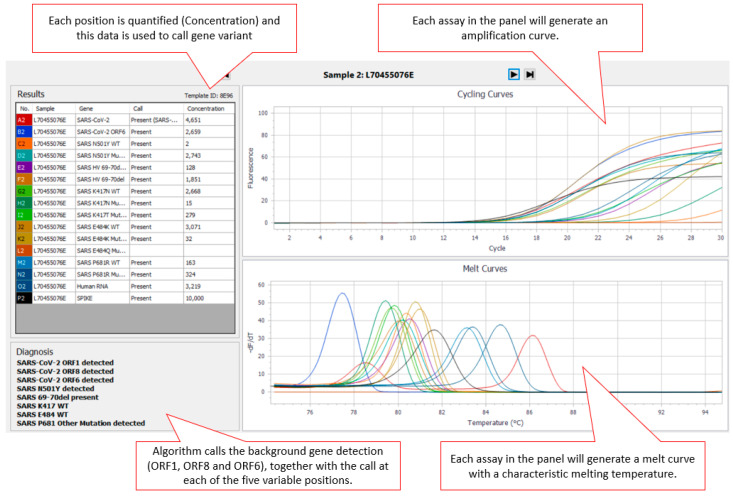
Screenshot of assay output using AusDiagnostics results software. The upper right panel shows the amplification curves for the individual assays, while the lower right panel shows their respective amplicon melt curve. The left panels show sample information and the output of the diagnostic algorithm.

**Figure 2 viruses-13-02028-f002:**
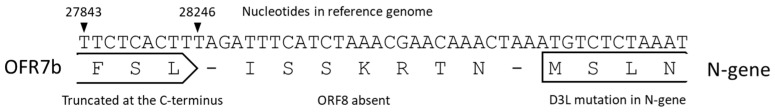
ORF8 deletion (Delta 394) observed in the cluster of samples from Oldham, UK.

**Table 1 viruses-13-02028-t001:** Sensitivity and specificity data.

Mutation	TP	FP	FN	TN	Sen.	Spec.
HV69/70 del	12	0	0	75	100% (95CI 69.8–100)	100% (95CI 93.9–100)
N501Y	16	0	0	71	100% (95CI 75.9–100)	100% (95CI 93.6–100)
E484K	5	0	0	82	100% (95CI 46.2–100)	100% (95CI 94.5–100)
K417N	4	0	0	83	100% (95CI 39.6–100)	100% (95CI 94.6–100)
P681R	7	0	0	5	100% (95CI 56–100)	100% (95CI 46.2–100)

TP—true positive (detected by Panel and WGS); FP—false positive (detected by panel but not by WGS); FN—false negative (not detected by Panel; but detected by WGS); and TN—true negative (not detected by either panel or WGS). Sensitivity: TP/(TP + FN) and Specificity: TN/(TN + FP) expressed as % together with 95% confidence intervals.

**Table 2 viruses-13-02028-t002:** Analytical sensitivity (limits of detection—LOD).

Variant	LOD
Reference WHO International Standard (NIBSC 20/146)	313 IU/mL
Alpha (VOC-20DEC-01, B.1.1.7)	49 copies/ml
Beta (VOC-20DEC-02, B.1.351	99 copies/mL
Gamma (VOC-21JAN-02, P.1)	197 copies/mL
Delta (VOC-21APR-02, B.1.617.2)	22 copies/mL

## Data Availability

Not applicable.

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
