# Peer review of "Development of a Multiplex Tandem PCR (MT-PCR) Assay for the Detection of Emerging SARS-CoV-2 Variants"

_viruses, 2021, doi:10.3390/v13102028_

Round 1

Reviewer 1 Report

I appreciate the effort of the authors but the methodological section is not clear enough. Please specificity a little bit more about how the multiplex is done. What primers are used?

It is not clear at all how can they relate the Ct curves with the genotype of the virus. They should be a little bit more clear about how can they reach these conclusions.

I think the idea is great but they are missing to explain key elements of how they get to the conclusions. It seems like the algorithm takes into account the melting temperature.  Is this right? If so what is the accuracy of this metric?

The article has great potential but they need to explain better what is going on so that it can be replicated. From this articles is practically impossible to replicate the results and that is not what science is about.

Author Response

We thank the reviewers for their helpful input and have amended the manuscript accordingly. Reviewer 1 asked for further clarity on the methods used, and Reviewer 2 asked for more detail in the introduction and background.

  1. Reviewer 1 has requested some clarification of the methods section to provide a better understanding of how the assay works, and how Ct related to genotype calling.  We acknowledge this point and have added an additional paragraph to the methods section to provide some clarification.
  2. Reviewer 1 has requested clarification on whether melt temperature is used in the calling algorithm. This is not the case, and a line has been added to emphasise this point.
  3. Reviewer 1 has stated that it would be difficult to replicate these results from this correspondence alone. We note this point but contend that the scope of the article is the validation of this specific assay, rather than the reporting of a new method – MT-PCR was first described in 2005 and has been available commercially since 2006.

The revised manuscript has been uploaded below with track changes selected.

Reviewer 2 Report

The study by Hale et al. in a communication form deals with the important development of a novel assay for detection of several variant of concerns (VOCs), namely the Alpha, Beta, Gamma and Delta variants. To my knowledge, the short version does not provide sufficient information at the introduction level about the problem here and in particular the structural and chemical character of the VOCs. Also, large part of the methodology can be placed in a supplementary material section. This is necessary to compare with other methods and understand the long-range effect of those mutations, and in particular why the author have chosen a reduced set of mutation for construction of the assay. For instance, the study employs 5 mutation sites: HV69/70, N501, K417, E484 and P681 and claim they are sufficient to detect “all” VOC. However, the delta variant possesses 2 key additional mutations the L452R and T478K which have not been discussed in the text. Indeed the P681R is present in the Indian lineage, but other variant such as the less transmissible Kappa does have it. I wonder how the study can differentiate between Kappa and Delta without including the L452 and T478 in the whole set.  Here, I recommend to discuss and implement if necessary those two mutation in the assay.

  In addition, I highly recommend to include a brief description of the spike (S) protein using a cartoon for indication of mutation positions and discuss whether they have improved the S stability (https://doi.org/10.1039/D0NR03969A), transmissibility and infection of SARS-CoV-2 (https://dx.doi.org/10.1371/journal.ppat.1009233)

Author Response

We thank the reviewers for their helpful input and have amended the manuscript accordingly. Reviewer 1 asked for further clarity on the methods used, and reviewer 2 asked for more detail in the introduction and background.

  1. Reviewer 2 has requested some additional background on the selection of mutations targeted by this panel. We agree with this point and have provided an additional paragraph in the Introduction section, together with a reference to the current UK Government requirements.
  2. Reviewer 2 has recommended that we include a brief description of the spike (S) protein using a cartoon for indication of mutation positions. The conformation of the spike trimer is actually quite complex and well beyond the scope of this article. However, we have added two references to recently published articles that provide a good background on Spike structure and function for the benefit of the reader.
  3. Reviewer 2 Asks how the panel can distinguish between Kappa and Delta. As these differ at the E484 position (E484Q vs E484 wt) the assay as described can discriminate between these and we don’t believe that this point needs to be specifically reemphasised. Reviewer 2 has asked why we do not specifically refer to L452R and T478K in the delta variant.  The design of this assay (as has now been emphasised in an additional paragraph in the introduction) has paid particular notice to the UK Government guidance in the selection of SNPs to target. Consequently, we do not think that additional discussion would be worthwhile on why individual SNPs from the many possible choices were included or excluded in the design. 

A revised manuscript has been uploaded below with track changes set

Round 2

Reviewer 2 Report

I consider there has not made a big effort to answer my previous comments and further explain the determination of Delta variant. As it seems to me they could use the technique to determine all possibles VOCs. They need to answer how the assay is limited to certain variants and no other.

For instance, take a look at statement “was designed to use a diagnostic algorithm to predict the genotype present at five positions in the SARS-CoV-2 sur- face glycoprotein (S gene) (501, 69-70, 417, 484, & 681) associated with SARS-CoV-2 VOC (Variants of Concern).” The Delta variant has a mutation in residue L452R which is not considered. So I wonder how they get it. 

As I recommended there is a need to describe the structure of the S protein including mutations and deletions. So the reader can correlated the structure with construction of the assay. 

Also, I do appreciate a brief physicochemical description of the S protein. To answer: stability of WT with respecto to new variants, role of the glycan shield that is related to cell recognition and high infection (https://doi.org/10.1039/D0NR03969A or https://doi.org/10.1021/acscentsci.0c01056). Issues that are relevant to comment and could potentially affect the assay.

Author Response

Point 1. I consider there has not made a big effort to answer my previous comments and further explain the determination of Delta variant. As it seems to me they could use the technique to determine all possibles VOCs. They need to answer how the assay is limited to certain variants and no other.

We have addressed this point by including an additional paragraph in the discussion section further explaining that the range of SNPs for this assay format can be extended beyond the current set, and restating that the SNPs chosen were done so to reflect the UK national guidance, to which a reference is provided.

Point 2. For instance, take a look at statement “was designed to use a diagnostic algorithm to predict the genotype present at five positions in the SARS-CoV-2 sur- face glycoprotein (S gene) (501, 69-70, 417, 484, & 681) associated with SARS-CoV-2 VOC (Variants of Concern).” The Delta variant has a mutation in residue L452R which is not considered. So I wonder how they get it.

The reviewer has correctly pointed out that the delta variant includes L452R. However, this mutation is not unique to this lineage (it also occurs in the kappa variant and the pango A.2.5.1, C.40 & B.1.429 lineages).

As I recommended there is a need to describe the structure of the S protein including mutations and deletions. So the reader can correlated the structure with construction of the assay.

We would, in theory, be able to include a simple linear  representation of the location of these SNPs within the spike protein (along the lines of this diagram from Wikipedia: https://en.wikipedia.org/wiki/File:SARS-CoV-2_Theta_variant.svg), but we do not believe that this would add anything useful to the readers understanding. These are simply polymorphisms within a gene that can be used to determine a genotype, and by association a variant.

Point 4. Also, I do appreciate a brief physicochemical description of the S protein. To answer: stability of WT with respecto to new variants, role of the glycan shield that is related to cell recognition and high infection (https://doi.org/10.1039/D0NR03969A or https://doi.org/10.1021/acscentsci.0c01056). Issues that are relevant to comment and could potentially affect the assay.

The relationship between Spike structure and function, and the role of individual polymorphisms is an important and interesting topic. However, we do not understand how this has any place in the development and validation of a PCR-based genotyping assay. We therefore do not think that further inclusions are necessary. As we have already provided some references to published work on Spike in the first revision we do not think that we can add anything else here.